# Appraisal of the Flow Diversion Effect Provided by Braided Intracranial Stents

**DOI:** 10.3390/jcm13123409

**Published:** 2024-06-11

**Authors:** Ferdi Çay, Anıl Arat

**Affiliations:** 1Department of Radiology, Hacettepe University School of Medicine, Hacettepe Mh., A.Adnan Saygun Cd., Ankara 06230, Türkiye; drferdicay@gmail.com; 2Department of Neurosurgery, Yale University School of Medicine, New Haven, CT 06520, USA

**Keywords:** intracranial aneurysm, stent, flow diversion, braided, laser cut

## Abstract

**Objective:** Comparison of the results of stent-assisted coiling (SAC) with braided stents (BS), flow diverters (FD), and laser-cut stents (LCS) to determine the relative flow-diverting capacity of BS (Leo baby and Accero). **Methods:** Saccular intracranial aneurysms treated by SAC and FD-assisted coiling were retrospectively evaluated. Aneurysm occlusion, as graded per Raymond–Roy score, was categorized as either recanalization/stable residual filling (Group A; lacking a flow diversion effect) or stable/progressive occlusion (Group B with a “flow diversion effect”). Factors predicting the flow diversion effect were evaluated. **Results:** Of the 194 aneurysms included, LCS, BS, and FD were used in 70 (36.1%), 86 (44.3%), and 38 (19.6%) aneurysms, respectively. Aneurysms treated by FD were larger, had wider necks, and were located on larger parent arteries (*p* < 0.01, 0.02, and <0.01, respectively). The mean imaging follow-up duration was 24.5 months. There were 29 (14.9%) aneurysms in Group A and 165 (85.1%) in Group B. Among a spectrum of variables, including sex, age, aneurysm size, neck width, parent artery diameter, follow-up duration, and stent type, the positive predictors for stable/progressive aneurysm occlusion were aneurysm size and placement of an FD or BS (*p* < 0.01 and *p* < 0.01, respectively, and were positive predictors over LCS: ORs 6.34 (95% CI: 1.62–24.76) and 3.11 (95% CI: 1.20–8.07), respectively) in multivariate analysis. **Conclusions:** The placement of BS was a predictor of flow diversion over laser-cut stents. However, the flow diversion effect was approximately half that of FDs, suggesting that BS may only be considered to have some (partial) flow diversion effects.

## 1. Introduction

The most feared complication of intracranial aneurysms (IAs) is subarachnoid hemorrhage, with an incidence of 6 to 9 per 100,000 people each year [1]. Initially, endovascular treatment with coils provided a robust alternative to neurosurgical treatment but was criticized for high recurrent rates. With the advancements in braiding and stent technology, especially the evolution of braided intracranial stents, the durability of endovascular treatment improved. Finally, with the introduction of flow diverters (which are actually densely woven intracranial braided stents), the recurrence rates of angiographically eliminated aneurysms decreased to negligible rates [2]. Although the mechanism of complete aneurysm occlusion obtained with stent-assisted coiling (SAC) is a topic of ongoing research [3], current evidence suggests that flow remodeling effect [1] leading to thrombosis and reduction in shear stress [4,5] are involved in the healing process of aneurysms.

Braided stents (BS) are suggested to have flow-diverting properties [5,6,7,8,9,10]. However, the presence of this effect has been disputed by some authors [11,12,13], and if such a flow diversion property exists at all, its extent has remained unknown to date [14]. In this study, we aimed to evaluate the relative flow diversion provided by BS by comparing stent-assisted coiling of aneurysms in which a BS, a laser-cut stent (LCS), or a flow diverter (FD) was utilized, and LCS-assisted coiling was taken as a reference.

## 2. Materials and Methods

Ethical approval was obtained from the institutional review board for this retrospective study. Consecutive patients with intracranial saccular aneurysms who underwent endovascular treatment with stent-assisted coiling were retrospectively evaluated. In this study, the open-cell Neuroform Atlas (Stryker, Kalamazoo, MI, USA) and the closed-cell Acclino (Acandis, Pforzheim, Germany) were used as LCS. Leo-baby (Balt, Montmorency, France) and Accero (Acandis, Pforzheim, Germany) were used as BS. Silk (Balt, Montmorency, France), Surpass (Stryker, Kalamazoo, MI, USA), and Derivo (Acandis, Pforzheim, Germany) families of devices were used as FDs. Intracranial stents differ according to their physical properties. The Neuroform Atlas stent is a self-expanding, hybrid but mainly open-cell design, nitinol stent with 3 radiopaque markers on each end, and it has no resheathability. The Atlas stent has one of the largest cell areas (up to 5 mm^2^) among all intracranial stents [15]. The Acclino stent is a self-expanding, closed-cell, nitinol stent with 3 radiopaque markers on each end, and it has 90% resheathability. The Acclino stent has an asymmetric cell design to improve vessel wall apposition [16]. The strut width and metal coverage of the Acclino stent are 0.3–0.7 mm and <10%, respectively. The Leo-baby stent is a self-expandable braided stent with 16 nitinol wires (2 of which are radiopaque), and it has up to 90–95% resheathability. The cell size of the Leo-baby stent is lesser than the Acclino (0.9 mm and 1.8 mm, respectively) and Atlas stents. This feature provides a tighter mesh, preventing coil protrusion during coiling, and may provide flow diversion to some extent. The potential disadvantage of this feature is the difficulty of re-catheterization through the stent struts when the coiling microcatheter backs out of the aneurysm [17]. The Accero stent is a nitinol braided stent with full-length visibility due to platinum–nitinol composite wires with 3 markers on each end. It has resheathability up to 95% [18]. The metal coverage of the Accero stent is 15–20%. Flow diverter stents are made of multiple braided wires ranging from 48 to 64, and their metal coverage ratio differs between 22% and 70% [19,20]. Surpass Evolve is a 48- to 64-wire flow diverter with a relatively constant mesh density and as such is similar to its predecessor, the Surpass Streamline, with a similar radial force [21]. The Surpass Evolve has a nominal metal coverage ratio of 22–26% and a pore density of 15–30 (pore/mm^2^). The Silk device is a braided flexible mesh stent made of 44 nitinol and four platinum wire strands with flared ends that act as radiopaque markers. The Silk Vista Mini is a lower-profile newer generation device that enables the usage of a flow diverter in distal arteries. The Derivo flow diverter consists of 48 nitinol wires with an inner platinum core, for enhanced visibility, and three radiopaque markers at each end. The porosity of the Derivo stent is 56–70%, respectively. The newer generation device, the Derivo Mini, has closed loops at its distal end. All of the above-listed flow diverters are resheathable [19,20].

Patients without follow-up, patients with ruptured aneurysms, and those with dissecting/fusiform aneurysms were excluded from the study. Because ruptured aneurysms are not morphologically stable and there is a subset of ruptured aneurysms enlarging substantially despite optimal endovascular therapy, the inclusion of these aneurysms could potentially act as a confounder for the evaluation of the treatment response, possibly lowering the treatment arm that harbors a higher ratio of such aneurysms. Therefore, ruptured aneurysms were excluded from the study. As the dissecting/fusiform aneurysms were primarily treated with flow diversion without coiling, we did not include patients with these aneurysms in the cohort as well. Patients were only included if only a single BS or LCS was used to assist coiling, and patients with SAC with dual stenting (Y- or X-stenting) were excluded from the study. The decision to proceed with an FD instead of a stent was based on anatomical factors, such as the location of the aneurysm, the availability of appropriately sized devices that match the arterial dimensions of a specific patient, and for bifurcation aneurysms, the expected ability to catheterize the acutely angled efferent branch and the possibility of bulging the stent into the aneurysm to protect the origin of the side branches. For ICA aneurysms, flow diversion without coiling was generally the initial modality. For bifurcation aneurysms, an initial attempt was made to treat the aneurysms without stent assistance. If this attempt failed, stent or FD-assisted coiling was utilized. Stent-assisted coiling was considered the initial strategy. In some cases, the initial assumption of the operator that stent-assisted coiling would yield an acceptable aneurysmal obliteration proved to be wrong after stent deployment, and then FDs were used together with the stent. SAC cases in which a stent and FD were placed telescopically were grouped among the flow-diverted SAC cases for the final analysis. Patient characteristics (age and sex), aneurysm characteristics (aneurysm size, neck width, proximal and distal diameter of the stented artery harboring the aneurysm, bifurcation/sidewall), type of stent used for SAC procedures (LCS, BS, or FD), follow-up duration, and aneurysm occlusion status immediately after the procedure and during the follow-up, based on the Raymond–Roy (RR) score, were analyzed. Posterior communicating artery aneurysms were classified as bifurcation aneurysms if the posterior cerebral artery was fetal. Otherwise, they were classified as sidewall aneurysms.

In the literature, progressive occlusion of a “significant proportion” of coiled aneurysms and diminished recanalization rates have been proposed as proxies for the flow diversion effect of a stent [9,14,22]. Consequently, in this study, to investigate the flow diverter effects of different types of stents, aneurysms were divided into two groups. Group A included aneurysms with recanalization (aneurysms with deterioration of the RR score during follow-up) or residual filling (aneurysms with an RR2 or an RR3 score at initial post-procedure angiogram that remained as such during the follow-up). Group B included aneurysms with stable occlusion (aneurysms with an RR1 score at the initial post-procedure angiogram that remained totally occluded during follow-up) or progressive occlusion (aneurysms with improvement in the RR score during follow-up). Examples of aneurysms included in both groups are provided in the Appendix A.

The SPSS 25.0 (IBM Corp, Armonk, NY, USA) program was used for statistical analysis. Continuous data are presented as the mean ± SD or median (IOR), and categorical data are presented as percentages. The distribution of continuous data was analyzed with the Shapiro–Wilk test. For the continuous variables with normal distribution, the independent sample *t*-test was used for comparison. For the continuous variable without normal distribution, the Mann‒Whitney U test was used for comparison. For the comparison of three independent groups, a one-way ANOVA test was used. For univariate analysis, categorical variables were compared with a chi-square test (Fisher’s exact test if required). Statistical significance was set at *p* < 0.05. Variables were entered in the multivariate analysis based on their statistical significance (*p* < 0.05) in the univariate analysis. Since the outcome was binary, multivariate analysis between groups A and B was performed with the use of multivariate logistic regression. 

## 3. Results

One hundred seventy-six patients, 109 (61.9%) female and 67 (38.1%) male, with 194 intracranial aneurysms were included in the study. Aneurysm locations are listed in Table 1. 

One hundred seventy-one aneurysms (88.1%) were bifurcation aneurysms. The types of stents that were used for SAC procedures are listed in Appendix A. For the stent type, the reference category was LCS. Of the 70 LCS, 61 (87.1%) were the open-cell Neuro-form Atlas stents and 9 (12.9%) were the closed-cell Acclino stents. As there was no statistically significant difference between these two stents based on initial (*p* = 0.646), final follow-up (*p* = 0.788), or stable/progressive (*p* = 0.673) occlusion rates, both stent types were included as a single LCS reference group.

The initial RR scores immediately after the procedure according to the stent type are listed in Appendix A. The aneurysm characteristics of the patients who underwent treatment with FDs were different from those treated with LCS and BS. They were larger in size and had wider necks, and the proximal and distal diameters of the parent artery were larger (Appendix A).

Six-month follow-up was available for 181 patients (93.3%), 174 (96.1%) with DSA, and 7 (3.9%) with MRA. Only 28 patients (14.4%) had less than 12 months of follow-up in this cohort, and the median follow-up time was 6 months (IOR: 3) for these 28 patients. One hundred sixty-six patients (85.6%) had 12 or more months of follow-up. Beyond 12 months, follow-up was performed with MRA in 141 patients (84.9%) and with DSA in 25 patients (15.1%).

There were 29 (14.9%) aneurysms in Group A and 165 (85.1%) aneurysms in Group B. In Group A, there were 14 (7.2%) aneurysms with recanalization and 15 (7.7%) aneurysms with ongoing residual filling. In Group B, there were 132 (68%) aneurysms with stable occlusion and 33 (17%) aneurysms with progressive occlusion (Appendix A). The univariate analysis of patient characteristics (age, sex), aneurysm characteristics (aneurysm size, neck width, proximal/distal diameter of the stented artery), follow-up time, and stent type between the two groups are listed in Table 2. Variables that reached statistical significance in univariate analysis, aneurysm size (*p* = 0.001) and neck size (*p* = 0.009), were entered in the multivariate analysis. Stent type was also included in the multivariate analysis, because of its clinical relevance and its *p*-value (0.06) was very close to statistical significance. Aneurysm size and stent type reached statistical significance in the multivariate logistic regression (Table 3). In the multivariate analysis, BS or FD was a positive predictor for stable/progressive occlusion over LCS. The odds ratio was higher for FDs than for BS (ORs were 6.34 (95% CI: 1.62–24.76) and 3.11 (95% CI: 1.20–8.07), respectively).

## 4. Discussion

Flow diversion has become an integral component of the endovascular treatment of IAs. The currently accepted range of efficacy for flow diversion was set by early flow diverter studies published approximately 10 years ago, which yielded a benchmark of 73.6–76% for total aneurysmal occlusion at 6 months [23]. It is known that “flow diversion”, as defined by these initial studies, can be attained with braided devices having a porosity of approximately 65%. On the other hand, the exact physical properties that define a “flow diverter” (e.g., number of struts, strut thickness, porosity, and pore density) are multiple, and flow diversion does not simply correspond to a vessel wall coverage of 35% [24]. In this sense, flow diversion may be regarded as a continuous rather than binary variable that results in sufficient intra-aneurysmal stasis after endovascular treatment [25]. Although an exact definition of flow diversion is lacking, it is clearly known that progressive occlusion and minimal/no recanalization [26,27,28] are the hallmarks of flow diverters; that is, IAs treated with FDs are expected to occlude progressively and then maintain their occlusion during the follow-up [2].

BS is believed by many interventionalists to have flow-diverting properties [6,7,8,29]. Some interventionalists further postulate that there is an overlap between the porosities of FDs and the newer generation BS, resulting in a “certain flow diverter effect” [30,31]. However, this belief is not supported by sound data. Although the results in the stent-assisted coiling arm of our cohort are more or less similar to these studies, we further included an FD-assisted group to support our assertion that coiling is a confounder related to total aneurysmal occlusion. Thus, the final occlusion rate obtained with BS-assisted coiling in these studies and in our study is not purely a result of flow diversion. There are animal data [32] and clinical data [9,13] that challenge the assumption of a “certain flow diverter effect” of BS. The flow-diverting capacity of standalone stents (or coils with or without stents for that matter [5,33,34,35]) has not been proven to date in the clinical setting. Instead, there are in vitro studies that compare the hemodynamic effects of FDs, BS, and LCS [5,35]. As flow diverters rely on physiological mechanisms [36], the results of these in vitro studies remain speculative only. There are also clinical publications that compare FDs with BS [37,38,39] or BS with LCS [40,41]. However, the efficacy comparison between the FDs and stents reported in these studies is based on the comparison of stent-assisted coiling with bare flow diverters (i.e., without coiling), precluding the evaluation of the positive contribution of coils to the final occlusion rate in aneurysms treated with SAC, potentially overstating the flow diversion effects of the stents.

Despite the abundant comparisons of FDs with BS or LCS in the literature [37,38,39,42,43,44], to be able to determine the level of flow diversion provided by BS specifically, the inclusion of an LCS (as a reference) arm, a BS arm, and an FD separately in the same cohort is crucial; otherwise, the extent of the flow diversion provided by BS would remain undisclosed. Such a comparison has not been performed previously, but it does have clinical relevance: Stent monotherapy and suboptimal coiling (undercoiling) have been proposed as BS-based endovascular treatment alternatives, despite the lack of clear evidence as to the flow diversion capacity of these stents [29,45,46,47]. Due to the uncertainty about what represents an undercoiled aneurysm and what the utility of stent monotherapy in the general population of intracranial aneurysms should be, there is a need for the determination of the flow diversion effects of stents, especially braided stents.

When we minimized the confounding effect of the coils (by including only coiled aneurysms) for a one-to-one comparison of the flow diversion effect of different stents, the stent type was a predictor of stable and persistent aneurysm occlusion (Group B), which is a proxy for the flow diversion effect. When the LSC was set as a reference, ORs were 3.11 (95% CI: 1.20–8.07) and 6.34 (95% CI: 1.62–24.76) for BS and FDs, respectively. These results place the flow diversion effect of the BS somewhere between those of LCS and FD and very likely account for the higher occlusion rates in SAC with BS over LCS in some series [40]. The fact that flow diversion provided by BS is somewhere between LCS and FD most likely stems from differences in porosity. The highly porous LSC, with pore sizes of several millimeters, is much higher than BS. In turn, BS pore sizes in the range of about 1 mm are higher than FD and lead to “incomplete” flow diversion. This incomplete diversion may additionally vary by a multitude of factors (such as deployment technique, sizing, and parent artery angulation). Thus, the operators should rely on the completeness of coiling and therefore make every effort to coil the aneurysms thoroughly, rather than trusting the varying degrees of flow diversion of BS. The fact that the long-term occlusion rates of SAC with BS did not exceed LCS in some comparative studies [11,12,13] may be related to the undercoiling of aneurysms in some centers with the expectation of flow diversion to take over with BS. An alternative explanation may be the dense coiling in these series to the level of complete occlusion initially, diminishing the FD effect of BS on the final occlusion rate [13,48]. Stable residual filling and recanalization were observed in bifurcation aneurysms treated with FD-assisted coiling (3 stable residual aneurysms and 1 recanalization) during long-term follow-up in this cohort. Since residual/recurrent aneurysm formation is possible even with FD-assisted coiling, as a precaution, practitioners should refrain from relying too much on the flow-diverting effect of BS during SAC; that is, undercoiling of an aneurysm should not be performed solely on the assumption that the residual aneurysm will be occluded during the follow-up by the flow diversion provided by the BS. One other precaution would be limiting the use of stent monotherapy in selected cases. Notwithstanding the current enthusiasm for monotherapy, the occlusion rates of this treatment for typical saccular aneurysms were at most 50% in recent studies [6,49], and higher occlusion rates were available only for selected fusiform and small saccular aneurysms [9]. In addition to the persistent risk of rupture, residual aneurysms after undercoiling or monotherapy pose a technical challenge, and reentry into the aneurysm sac through BS may be cumbersome [50].

One other unique feature of our study is that in previous publications that compared BS with FD, a single type of BS (LVIS/LVIS Jr, Microvention, Tustin, CA, USA) was used. A comparative study of the flow diversion between FD and BS has not been performed with the remaining two devices reported in our study. In this respect, our study fills another knowledge gap in the literature.

In our study, we compared LSC, BS, and FD in coiled aneurysms. In addition to its retrospective methodology, this is a major limitation of this study. It would be ideal to compare FD without coiling to BS/LCS without coiling, but due to the current standards of endovascular practice, short of the cases treated by staged embolization or the few cases of stent monotherapy, stenting without coiling is not performed commonly enough to provide a sound comparison cohort. This is why we had to compare coiled aneurysms treated with FD to those treated with BS so that we could delineate the extent of flow diversion provided solely by BS by canceling out the effect of coiling. One other limitation is the selection bias. Flow diverters were used if BS or LCS-assisted coiling had an unsatisfactory result. Selection bias may have potentially affected the results of this study. Moreover, there may be differences between occlusion rates between the subtypes (open versus closed-cell stents) of LCS [51], which may have been missed in our cohort due to the low number of Acclino stents used in this study. 

Finally, due to the variability of coils and also the coiling technique, as well as the lack of a definition of an “acceptable occlusion” of an aneurysm with coils, our single-center results may not be generalizable to other centers. A larger dataset from multiple centers may overcome this limitation and our limitation related to the relatively small size of our cohort. More ideally, a multicenter study comparing aneurysms treated by single-flow diverter placement and sole stenting and longer duration of follow-up (given the longer time needed to display the full potential for flow diversion) will provide more reliable results about the extent of flow diversion provided by BS.

## 5. Conclusions

In this study, there was an approximately three times higher rate of stable or improved aneurysm occlusion after stent-assisted coiling with BS compared to LCS. In turn, this rate was approximately half the value attained by FD-assisted coiling, suggesting that BS may be considered to have merely some (partial) flow diversion effect. It is advisable that during BS-assisted coiling, the aneurysms should be coiled as safely but, as densely as possible in the meantime.

## Figures and Tables

**Table 1 jcm-13-03409-t001:** Aneurysm locations.

Locations	*n* (%)
AChA	2 (1)
ACOM	46 (23.7)
AICA	2 (1)
Basilar tip	8 (4.1)
ACA	8 (4.1)
MCA	88 (45.4)
PCA	4 (2)
PCOM	5 (2.6)
SCA	4 (2.1)
Supraclinoid ICA	12 (6.2)
Terminal ICA	13 (6.7)
PICA	2 (1)

ACA: anterior cerebral artery, AChA: anterior choroidal artery, ACOM: anterior communicating artery, AICA: anterior inferior cerebellar artery, ICA: internal carotid artery, MCA: middle cerebral artery, PCA: posterior cerebral artery, PCOM: posterior communicating artery, SCA: superior cerebellar artery.

**Table 2 jcm-13-03409-t002:** Univariate analysis of the variables between Groups A and B.

Variables	Recanalization + Stable Residual Filling (Group A), *n* = 29	Stable Occlusion + Progressive Occlusion (Group B), *n* = 165	*p* Value
Female gender (%)	19 (65.5)	102 (61.8)	0.70
Age, mean ± SD	50.59 ± 13.78	53.63 ± 12.58	0.34
Aneurysm size (mm), mean ± SD	9.96 ± 3.31	7.21 ± 3.77	<0.01 *
Neck width (mm), mean ± SD	4.91 ± 1.41	4.08 ± 1.52	<0.01 *
Proximal diameter of the stented artery (mm), mean ± SD	2.76 ± 0.69	2.63 ± 0.63	0.39
Distal diameter of the stented artery (mm), mean ± SD	2.02 ± 0.52	1.98 ± 0.59	0.61
Follow-up time (months), mean ± SD	22.52 ± 11.85	24.84 ± 15.75	0.76
Stent type (%) Laser cut Braided Flow diverter	16 (55.2) 9 (31) 4 (13.8)	54 (32.7) 77 (46.7) 34 (20.6)	0.06 *




SD: standard deviation, * entered in multivariate logistic regression.

**Table 3 jcm-13-03409-t003:** Multivariate logistic regression analysis.

Covariate	B	*p* Value	OR	95% CI	
				Lower	Upper
Stent type Braided over laser cut Flow diverter over laser cut	1.13 1.84	0.009 0.019 0.008	3.11 6.34	1.20 1.62	8.07 24.76
Aneurysm size	−0.22	<0.01	0.79	0.71	0.89
Neck width	−0.16	0.404	0.85	0.58	1.24

OR: odds ratio, CI: confidence interval.

## Data Availability

The data presented in this study are available on request from the corresponding author.

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
