# Peer review of "Appraisal of the Flow Diversion Effect Provided by Braided Intracranial Stents"

_jcm, 2024, doi:10.3390/jcm13123409_

Round 1
Reviewer 1 Report
Comments and Suggestions for Authors
This is a well written retrospective account related to lasting aneurysm occlusion comparing flow diverters, braided stents and laser cut stents. Braided stents performed well, suggesting a good flow diverting effect. The material is well presented and of substantial interest.
The description refers solely to the generic types of implants. The brand names and manufacturer should be given, since differences may be present among different models.
Author Response
Comment 1: This is a well written retrospective account related to lasting aneurysm occlusion comparing flow diverters, braided stents and laser cut stents. Braided stents performed well, suggesting a good flow diverting effect. The material is well presented and of substantial interest.
The description refers solely to the generic types of implants. The brand names and manufacturer should be given, since differences may be present among different models.
Reply: We appreciate the reviewer’s comment. The following sentences related to the brand names of stents were added to the materials and methods section:
“In this study, Neuroform Atlas (Stryker, Kalamazoo, MI) and Acclino (Acandis, Pforzheim, Germany) were used as LCS. Leo-baby (Balt, Montmorency, France) and Accero (Acandis, Pforzheim, Germany) were used as BS. Silk (Balt, Montmorency, France), Surpass (Stryker, Kalamazoo, MI) and Derivo (Acandis, Pforzheim, Germany) families of devices were used as FD.” Please see Reply to Comment 1 under Reply to the Editor for further description of these devices.
Reviewer 2 Report
Comments and Suggestions for Authors
1. In the opening sentences of the Introduction section, consider adding statistics regarding the prevalence or rupture risk of aneurysms, as well as the significance of stenting. This will underscore the importance and relevance of your project for the audience.
2. The introduction lacks a literature review. Initially, you should provide a comprehensive review of previous papers and gradually guide the readers to the gap you wish to fill. Then, you must emphasize the parameters you intend to use to support your hypothesis. For example, particularly, you should mention the complexities associated with small aneurysms and their risk of rupture [https://doi.org/10.1007/s10143-020-01367-3]. You should also discuss the pros and cons of coils and stents [https://doi.org/10.1179/1743132814Y.0000000318] [https://doi.org/10.1136/pgmj.2010.105387]. Ultimately, you should present an overview of the effect of various variables that you considered, as well as those not considered in previous studies to assess the effects of flow diversion. One of the most important parameters is blood hemodynamic parameters (specially wall shear stress), which previous studies have highlighted [https://doi.org/10.1134/S0021894417060025], along with the morphology of the aneurysm. Including these sections and related references is essential to enhance the readers' understanding of the paper. However, currently, you abruptly transition to your main goal. Additionally, this introduction does not meet the standards of an introduction.
3. Is the hypothesis of this study clear?
4. The rationale for excluding patients with ruptured aneurysms and those with dissecting/fusiform aneurysms should be discussed in terms of how these conditions could affect the results. Why? Possible biase? ..
5. The paper should ensure that all necessary assumptions for the statistical tests employed (like normality for t-tests) are verified and reported. It could be helpful to include a more detailed discussion on the selection of variables included in the multivariate model, particularly why some variables were included over others based on their p-values from univariate analysis.
6. Explain in detail which multivariate method you used. Why does the second part of Table 2 list only three parameters? Where are the other parameters in your multivariate analysis? Did you encounter a multicollinearity problem? If so, how did you handle it? Report the VIF scores? Did you use a linear or polynomial multivariate method, and why? Did you use a correction method? If so, which one? Could you provide the ROC curves? Report the results of the overfitting and underfitting analysis. Share the results of the cross-validation analysis?
7. The paper concludes that braided stents (BS) have a "semi-flow diverting" effect compared to laser-cut stents (LCS) and flow diverters (FD). WHY? Explore the clinical reason of this finding more deeply. How do these results translate into clinical practice, especially considering the varying degrees of flow diversion?
8. The findings suggest that BS are effective but less so than FDs. WHY? The discussion could benefit from a clearer explanation of the practical implications for clinical decision-making. Are BS recommended only in specific scenarios where FDs are not suitable?
9. Comparison of your results with corresponding results of previous studies is the main concern about the Discussion section. The paper could also further discuss contrasting studies or conflicting evidence to provide a balanced view of the current understanding of flow diversion in intracranial aneurysms. It may also be beneficial to discuss newer developments in stent technology and how these might impact the results of future studies or clinical practices.
10. The study is based on a single-center dataset, which may limit the generalizability of the findings. It could be useful to discuss how the findings might differ in other settings or populations. Considering the retrospective nature and the specific patient cohort studied, a discussion on how these findings can be applied to the broader patient population would be beneficial.
11.Limitations of this study should explain in detail with possible suggestions for future studies.
12. The English language level of this paper needs major revisisons. Ensure consistent terminology and abbreviation use throughout the paper to avoid confusion (e.g., BS, LCS, FD should be consistently defined at first use).
Comments on the Quality of English LanguageModerate
Author Response
Comment 1: In the opening sentences of the Introduction section, consider adding statistics regarding the prevalence or rupture risk of aneurysms, as well as the significance of stenting. This will underscore the importance and relevance of your project for the audience.
Reply: The following paragraph was added to the beginning of the introduction section.
“The most feared complication of intracranial aneurysms is subarachnoid hemorrhage, with an incidence of 6 to 9 per 100.000 people each year [1]. Initially, endovascular treatment with coils provided a robust alternative to neurosurgical treatment but was criticized for high recurrent rates. With the advancements in braiding and stent technology, especially the evolution of braided intracranial stents, the durability of endovascular treatment improved. Finally, with the introduction of flow diverters (which are actually densely woven intracranial braided stents) the recurrence rates of angiographically eliminated aneurysms decreased to negligible rates [2].”
Comment 2: The introduction lacks a literature review. Initially, you should provide a comprehensive review of previous papers and gradually guide the readers to the gap you wish to fill. Then, you must emphasize the parameters you intend to use to support your hypothesis. For example, particularly, you should mention the complexities associated with small aneurysms and their risk of rupture [https://doi.org/10.1007/s10143-020-01367-3]. You should also discuss the pros and cons of coils and stents [https://doi.org/10.1179/1743132814Y.0000000318] [https://doi.org/10.1136/pgmj.2010.105387]. Ultimately, you should present an overview of the effect of various variables that you considered, as well as those not considered in previous studies to assess the effects of flow diversion. One of the most important parameters is blood hemodynamic parameters (specially wall shear stress), which previous studies have highlighted [https://doi.org/10.1134/S0021894417060025], along with the morphology of the aneurysm. Including these sections and related references is essential to enhance the readers' understanding of the paper. However, currently, you abruptly transition to your main goal. Additionally, this introduction does not meet the standards of an introduction.
Reply: We evaluated this comment together with the relevant comment of the editor. The following sentences and references were added to the Introduction section. “Although the mechanism of complete aneurysm occlusion obtained with stent-assisted coiling is a topic of ongoing research (https://doi.org/10.1136/pgmj.2010.105387), current evidence suggests that flow remodeling effect (https://doi.org/10.1179/1743132814Y.0000000318) leading to thrombosis and reduction of shear stress (https://doi.org/10.1134/S0021894417060025) (https://doi.org/10.1186/s12967-016-0959-9) are involved in the healing process of aneurysms.”
Comment 3: Is the hypothesis of this study clear?
Reply: Our hypothesis in this study was “Is the stent type used for stent-assisted coiling an independent predictor of stable or progressive occlusion (a proxy for the flow diversion effect of the stent)?” If it was, we wanted to determine the extent of the flow diversion induced by the stent. The study results showed that braided and flow-diverter stents were predictive of stable or progressive occlusion over laser-cut stents. Three times higher rates of stable or progressive occlusion were achieved with braided stent-assisted coiling compared to the laser-cut sent-assisted coiling.
Comment 4: The rationale for excluding patients with ruptured aneurysms and those with dissecting/fusiform aneurysms should be discussed in terms of how these conditions could affect the results. Why? Possible biase? ..
Reply: The following was added to the Materials and Methods Section of the manuscript: “Because ruptured aneurysms are not morphologically stable and there is a subset of ruptured aneurysms enlarging substantially despite optimal endovascular therapy, the inclusion of these aneurysms could potentially act as a confounder for the evaluation of the treatment response, possibly lowering the treatment arm that harbors a higher ratio of such aneurysms. Therefore, ruptured aneurysms were excluded from the study. As the dissecting/fusiform aneurysms were primarily treated with flow diversion without coiling, we did not include patients with these aneurysms in the cohort as well.”
Comments 5 and 6: The paper should ensure that all necessary assumptions for the statistical tests employed (like normality for t-tests) are verified and reported. It could be helpful to include a more detailed discussion on the selection of variables included in the multivariate model, particularly why some variables were included over others based on their p-values from univariate analysis.
Explain in detail which multivariate method you used. Why does the second part of Table 2 list only three parameters? Where are the other parameters in your multivariate analysis? Did you encounter a multicollinearity problem? If so, how did you handle it? Report the VIF scores? Did you use a linear or polynomial multivariate method, and why? Did you use a correction method? If so, which one? Could you provide the ROC curves? Report the results of the overfitting and underfitting analysis. Share the results of the cross-validation analysis.
Reply: We consulted a second statistician for the analysis. He concurred that the distribution of The distribution of continuous data was analyzed with the Shapiro-Wilk test. For the continuous variables with normal distribution, the independent sample t-test was used for comparison. For the continuous variable without normal distribution, the Mann‒Whitney U test was used for comparison. For the comparison of three independent groups, a one-way ANOVA test was used. For univariate analysis, categorical variables were compared with a chi-square test (Fisher’s exact test if required). Statistical significance was set at p<0.05. Variables were entered in the multivariate analysis based on their statistical significance (p<0.05) in the univariate analysis. Since the outcome was binary, multivariate analysis between groups A and B was performed with the use of multivariate logistic regression.” These sentences were also added to the materials and method section.
Since the current imaging technology is unreliable to provide a numeric outcome for the occlusion assessment (such as a definite percent of aneurysm occlusion), we are unable to use linear or polynomial multivariate analysis. As our dependent variable was binary, recanalization/stable residual filling (Group A; lacking a flow diversion effect) AND stable/progressive occlusion (Group B with a “flow diverter effect”), we used multiple logistic regression. Hence, because our model was simple and only contained two categorical variables overfitting or underfitting was not observed and a correction method was not necessary. To our knowledge, cross-validation is unavailable for logistic regression analysis in the SPSS program.
As suggested by the reviewer the following explanation was included in the Material and Methods, and results section. “Variables were entered in the multivariable logistic regression based on their statistical significance (p< 0.05) in univariate analysis. Only, aneurysm size (p-value: 0.001) and neck size (p-value 0.009) were significant in univariate analysis. Stent type was also included in the multivariate logistic regression, because of its clinical relevance and its p-value (0.06) was very close to statistical significance.” To further clarify this, the results from multivariate logistic regression were taken out of Table 2 and a new table -Table 3- was created.
Multicollinearity evaluation was performed with the use of ado and DDAG packages in R (version 4.3.2).
VIF values were calculated for aneurysm size, neck width, and stent type for multicollinearity check. The VIF values were 0.774, 1.838, and 1.771, respectively, which were under ten. As a second method, three variables were first included in the model together and the variables were removed from the model one by one and their effects on the statistics of other variables were examined. It was observed that removing variables did not cause major changes in the statistics of other variables. Since we noted that multicollinearity could not be mentioned with either method, multivariate analysis was performed by including three variables together.
Finally, as there was no statistically significant difference in results between the two statistical calculations a change in the results was not needed.
Comments 7-8: The paper concludes that braided stents (BS) have a "semi-flow diverting" effect compared to laser-cut stents (LCS) and flow diverters (FD). WHY? Explore the clinical reason of this finding more deeply. How do these results translate into clinical practice, especially considering the varying degrees of flow diversion?
The findings suggest that BS are effective but less so than FDs. WHY? The discussion could benefit from a clearer explanation of the practical implications for clinical decision-making. Are BS recommended only in specific scenarios where FDs are not suitable?
Reply: Indeed, our results suggest that FDs may replace BS in certain scenarios. We added to following sentences in the discussion section to emphasize this point.
“The fact that flow diversion provided by BS is somewhere between LCS and FD most likely stems from differences in porosity. The highly porous LSC, with pore sizes of several millimeters, is much higher than BS. In turn, BS pore sizes in the range of about 1 millimeter are higher than FD and leads to “incomplete” flow diversion. This incomplete diversion may additionally vary by a multitude of factors (such as deployment technique, sizing, and parent artery angulation). Thus, the operators should rely on the completeness of coiling and therefore make every effort to coil the aneurysms thoroughly, rather than trusting the varying degrees of flow diversion of BS.”
“One other conclusion suggested by our results is that flow diverters may be utilized instead of BS in the endovascular treatment of cerebral aneurysms which are not likely to be completely (or almost completely coiled) as long as a valid surgical option does not exist.”
Comment 9: Comparison of your results with corresponding results of previous studies is the main concern about the Discussion section. The paper could also further discuss contrasting studies or conflicting evidence to provide a balanced view of the current understanding of flow diversion in intracranial aneurysms. It may also be beneficial to discuss newer developments in stent technology and how these might impact the results of future studies or clinical practices.
Reply: The following was included in the Discussion section: “Although the results in the stent-assisted coiling arm of our cohort are more or less similar to these studies, we further included an FD-assisted group to support our assertion that coiling is a confounder related to total aneurysmal occlusion. Thus, the final occlusion rate obtained with BS-assisted coiling in these studies and in our study is not purely a result of flow diversion.”
Comments 10-11: The study is based on a single-center dataset, which may limit the generalizability of the findings. It could be useful to discuss how the findings might differ in other settings or populations. Considering the retrospective nature and the specific patient cohort studied, a discussion on how these findings can be applied to the broader patient population would be beneficial.
Limitations of this study should explain in detail with possible suggestions for future studies.
Reply: This comment was evaluated together with the comment of reviewer 3 and the following phrases were added to the limitations section of the study.
“Finally, due to the variability of coils and also the coiling technique, as well as the lack of a definition of an “acceptable occlusion” of an aneurysm with coils, our single-center results may not be generalizable to other centers. A larger dataset from multiple centers may overcome this limitation and our limitation related to the relatively small size of our cohort. More ideally, a multicenter study comparing aneurysms treated by single flow diverter placement and sole stenting and longer duration of follow-up (given the longer time needed to display the full potential for flow diversion) will provide more reliable results about the extent of flow diversion provided by BS.”
Comment 12: The English language level of this paper needs major revisions. Ensure consistent terminology and abbreviation use throughout the paper to avoid confusion (e.g., BS, LCS, FD should be consistently defined at first use).
Reply: The manuscript was edited for a second time for language. Abbreviations were corrected as suggested.
Reviewer 3 Report
Comments and Suggestions for Authors
Dear author, Thank you for sending this paper for a review.
After a careful revision, I have some suggestions:
- English needs a revision;
- I appreciate that the authors reported the limitations of the study. The reported only the retrospective study. On the other hand, also the size of the sample is small/moderate(compared with the literature is small), the median follow-up is 2 years (compared with the literature, it is not the best).
- any abbreviation form needs a full form. (RR2 RR3 for example);
- some imaging or case presentation could improve the relevance for the readers.
Comments on the Quality of English Language
Grammar and syntax needs improvement
Author Response
Comment 1: English needs a revision
Reply: The manuscript was edited for a second time for language.
Comment 2: I appreciate that the authors reported the limitations of the study. The reported only the retrospective study. On the other hand, also the size of the sample is small/moderate(compared with the literature is small), the median follow-up is 2 years (compared with the literature, it is not the best).
Reply: This comment was evaluated together with the comments 10/11 of reviewer 2 and the following phrases were added to the limitations section of the study.
“Finally, due to the variability of coils and also the coiling technique, as well as the lack of a definition of an “acceptable occlusion” of an aneurysm with coils, our single-center results may not be generalizable to other centers. A larger dataset from multiple centers may overcome this limitation and our limitation related to the relatively small size of our cohort. More ideally, a multicenter study comparing aneurysms treated by single flow diverter placement and sole stenting and longer duration of follow-up (given the longer time needed to display the full potential for flow diversion) will provide more reliable results about the extent of flow diversion provided by BS.”
Comment 3: Any abbreviation form needs a full form. (RR2 RR3 for example)
Reply: Abbreviations were corrected as suggested and a full form was used on initial notation.
Comment 4: Some imaging or case presentation could improve the relevance for the readers.
Reply: We will be more than happy to provide more case presentations, in addition to those provided in the supplemental file, based on the preferences of the editor as noted below (editor comment 3).
Round 2
Reviewer 2 Report
Comments and Suggestions for Authors
Accept
Comments on the Quality of English LanguageAcceptable
Author Response
Thank you very much for helping us in improving our manuscript.
Reviewer 3 Report
Comments and Suggestions for Authors
Good improvement
Author Response

(The authors gave the same response as above.)
